# Zig-Zag Based Single-Pass Connected Components Analysis

**DOI:** 10.3390/jimaging5040045

**Published:** 2019-04-06

**Authors:** Donald G. Bailey, Michael J. Klaiber

**Affiliations:** 1Department of Mechanical and Electrical Engineering, School of Food and Advanced Technology, Massey University, Palmerston North 4442, New Zealand; 2Independent Researcher, 70176 Stuttgart, Germany

**Keywords:** connected components analysis, stream processing, feature extraction, zig-zag scan, hardware architecture, FPGA, pipeline

## Abstract

Single-pass connected components analysis (CCA) algorithms suffer from a time overhead to resolve labels at the end of each image row. This work demonstrates how this overhead can be eliminated by replacing the conventional raster scan by a zig-zag scan. This enables chains of labels to be correctly resolved while processing the next image row. The effect is faster processing in the worst case with no end of row overheads. CCA hardware architectures using the novel algorithm proposed in this paper are, therefore, able to process images at higher throughput than other state-of-the-art methods while reducing the hardware requirements. The latency introduced by the conversion from raster scan to zig-zag scan is compensated for by a new method of detecting object completion, which enables the feature vector for completed connected components to be output at the earliest possible opportunity.

## 1. Introduction

Connected components labelling is an important step in many image analysis and image processing algorithms. It processes a binary input image, for example after segmentation, and provides as output a labelled image where each distinct group of connected pixels has a single unique label. There are many different labelling algorithms (see for example the recent review [1]). Three main classes of algorithms are:Contour tracing [2,3], where the image is scanned until an object pixel is encountered. The boundary is then traced and marked, enabling all pixels to be labelled with the same label when scanning resumes.Label propagation algorithms [4] where labels are propagated through multiple passes through the image.Two pass algorithms, generally based on Rosenfeld and Pfaltz’s algorithm [5]. The first pass propagates provisional labels to object pixels from adjacent pixels that have already been processed. Sets of equivalent labels are processed to derive a representative label for the connected component, usually using some form of union-find algorithm [1,6]. Finally, the image is relabelled in a second pass, changing the provisional label for each pixel to the representative label.

The different two-pass algorithms fall into three broad classes: those that process single pixels at a time (e.g., [7,8]), those that process a run of pixels at a time (e.g., [9,10]), and those that process a block of pixels at a time (1 × 2 block in [11,12], 1 × 3 block in [13], and 2 × 2 block in [14,15]). There have been several FPGA implementations of connected components labelling (e.g., [16,17]), but the key disadvantage of these two-pass algorithms is the requirement to buffer the complete image between passes.

Connected component labelling is often followed by an analysis step, where a feature vector (usually based on the shape, but can also be based on statistics of the original image pixel values) is derived for each label. These feature vectors can then be used for subsequent classification, or even directly provide output data for some image analysis applications. When the labelling and feature vector measurement are combined as a single operation, it is termed connected components analysis (CCA).

Single pass CCA algorithms, introduced by Bailey and Johnston [18,19], extract feature data for each component during the initial provisional labelling pass. The labelled image, as an intermediate data structure, is no longer required, so the second relabelling pass can be skipped, enabling the complete algorithm to operate in a single pass. This has led to efficient low-latency hardware architectures that are able to operate directly on a video stream. The basic architecture of Figure 1 works as follows: For each pixel in the input stream, provisional labels are propagated from already processed pixels (represented by the neighbourhood window). Labels assigned in the current row are cached in a row buffer to provide the neighbourhood when processing the next row. When components merge, the associated labels are equivalent. One label is selected as the representative label (usually the label that was assigned the earliest), with the equivalence between the labels recorded in the merger table. Provisional labels saved in the row buffer may have been updated as a result of subsequent mergers and may no longer be current, so the output from the row buffer is looked up in the merger table to obtain the current representative label for the neighbourhood. For a single-pass operation, feature data is accumulated for each component on-the-fly within the data table. When components merge, the associated feature data is also merged. The component data is available after the component is completed, that is after no pixels extend from that component onto the current row.

The main limitation of the first single-pass algorithm [18] was that the data was only available at the end of the frame. In the worst case, this required resources proportional to the area of the image, preventing the use of on-chip memory for all but small images, or a restricted subset of images with a limited number of components. This was solved by Ma et al. [20], by recycling labels which requires identifying completed components, and freeing up the resources. Ma’s approach aggressively relabelled each component starting from the left of each row. It, therefore, required two lookups, one to resolve mergers, and one to translate labels from the previous row to the current row.

The next improvement in this class of CCA algorithms was developed by Klaiber et al. [21]. This solved the problem of two lookups by introducing augmented labels. Labels are allocated from a pool of recycled labels, and are augmented with row number to enable correct precedence to be determined when merging.

Trein et al. [22] took an alternative approach to single-pass CCA on FPGA, and run-length encoded the binary image first. Then, each run was processed in a single clock cycle, enabling acceleration when processing typical objects. In the worst case, however, the performance of run-based processing is the same as for pixel-based processing. Trein et al.’s method also suffers from the problem of chaining, although this was not identified in their paper.

The main issue with managing mergers on-the-fly is sequences of mergers requiring multiple look-ups to identify the representative label of their connected component. Those labels that require more than one lookup to lead to their representative label are referred to as *stale labels* [6]. This can occur after two or more mergers, where a single lookup in the merger table is insufficient to determine the representative label. Bailey and Johnston [18] identified chains of mergers that occur when the rightmost branch of a sequence of mergers is selected as the representative label (as illustrated in Figure 2). Before processing the next row, it is necessary to unlink such chains so that each old label directly points to the representative label. This unlinking is called path compression in union-find parlance.

The labels within such chains cannot occur later in the row because the label that was allocated the earliest was selected as the representative label. therefore, chain unlinking can be deferred until the end of each row [18]. Since the representative label within such a chain is rightmost, potential chain links can be saved on a stack enabling them to be unlinked from right to left. A disadvantage of such unlinking is that it incurs overhead at the end of each row. Typically, this overhead is about 1% [18], although in the worst case is 50% for a single row, or 20% for a whole image. A further complicating factor is that the overhead is image-dependent, and cannot be predicted in advance.

To overcome the chaining problem, Jeong et al. [23] proposed to directly replace all old entries within the row buffer with the new representative label whenever a merger occurs. This removes the unlinking overhead, and also the need for the merger table. To accomplish this, the row buffer must instead be implemented as a shift register, with each stage having a comparator to detect the old label, and a multiplexer to replace it with the representative label. Since such a content addressable memory cannot easily be implemented using a block memory, the resulting logic requires considerable FPGA resources.

Zhao et al. [24] also used aggressive relabelling, similar to Ma et al. [20], but instead used pixels as the processing unit, and runs as the labelling unit. The goal of this approach is to eliminate unnecessary mergers, and avoid the overhead at the end of each row. While labelling a run at a time does significantly reduce the number of mergers required, it does not eliminate chains of mergers (the pattern is more complex than Figure 2 of course). So although Zhao et al. claim to eliminate the end-of-row processing, without correctly resolving such chains, the results for some images will be incorrect.

Finally, Tang et al. [25] optimise this approach of using runs as a labelling unit to actually eliminate the end of row processing. They assign a unique label to each run, and rather than relabel runs when they connect, the connectivity is maintained within a linked list structure for each image row. The head of the list maintains the feature vector, and whenever a run is added to the list, both the list and data are updated. Clever use of the pointers enables the pointers to be kept in order, and enable the data to be accessed with two lookups, completely avoiding the problems with chains. It also means that labels are automatically recycled, and completed components are detected with a latency of one image row. There are two limitations of this algorithm: (1) It only handles 4-connectivity, rather than 8-connectivity which is usually used; Tang et al. also propose a pre-filter to convert an 8-connected image into the required 4-connected image prior to CCA. However, the pre-filter also means that incorrect values are derived for some features (e.g., area) without additional processing, although that processing is straight forward. (2) The outermost border of the image must be set to the background before processing; Tang et al. suggest extending the image with background pixels prior to processing to guarantee this condition. However, this would reintroduce 2 clock cycles per row overhead.

The primary contributions of this paper are: a novel approach to eliminate the end-of-row overhead associated with unchaining; and a novel method to detect completed components as soon as they are completed, giving a reduction in latency. These are based on a zig-zag based scan pattern through the image, with the algorithm outlined in Section 2. An FPGA architecture for realising zig-zag based CCA is described in detail in Section 3. The algorithm and architecture are analysed in Section 4 to show correct behaviour. Finally, Section 5 compares the new algorithm with existing single-pass pixel-based approaches.

## 2. Proposed Approach

Unchaining within the traditional algorithms [6,18,20,21] is effectively accomplished by performing a reverse scan back through the labels merged in the current row at the end of each row. This approach comes at the cost of having to introduce additional overhead to store the sequences of mergers in a stack data structure and unchain them sequentially at the end of each image row.

This paper proposes replacing the raster scan with a zig-zag scan, with every second row processed in the reverse direction. This enables chains of mergers to be resolved on-the-fly, as part of the merger table lookup and update process. The basic architecture of Figure 1 needs to be modified for the zig-zag scan, giving the system architecture of Figure 3. Although many of the blocks have the same name and function as those in Figure 1, the detailed implementation of many of these is changed.

First, a zig-zag reordering buffer is required in the input, to present the pixel stream in zig-zag order to the CCA unit. The row buffer also has to be modified to buffer data in zig-zag form. (Note that if the image is streamed from memory, this is unnecessary, as the pixels can directly be read from memory in zig-zag order.) Label selection is unchanged, as is the data table processing (apart from a novel extension to enable completed components to be detected earlier). The key changes are in the merger table processing for forming the neighbourhood, and merger control blocks. Zig-zag CCA is represented algorithmically in Algorithm 1. The nested **for** loops perform a zig-zag scan through the binary input image, with key steps as sub-algorithms described in the following sections.

### 2.1. Definitions

We first offer some definitions. The already processed pixels in the neighbourhood of the current pixel, *X*, are denoted *A*, *B*, *C*, and *D* as indicated in Figure 4. The labels associated with the neighbourhood pixels are designated $L_{A}$ through $L_{D}$. Background pixels are assigned label 0. A logic test of $L_{p}$ evaluates to true if pixel *p* is an object pixel and false if it is part of the background.

**Algorithm 1** Zig-zag CCA algorithm**Input:** Binary image *I* of width *W* and height *H***Output:** A feature vector for each connected component in *I*
1:StartOfLine:=False2:**for**y:=0**to**H−1**do**3:   **for**
x:=0
**to**
W−1
**when** y is even **else**
x:=W−1
**downto** 0 **do** ▹ Zig-zag scan4:     **if**
StartOfLine
**then**5:        ReverseNeighbourhood                     ▹ Algorithm 36:        StartOfLine:=False7:     **else**8:        UpdateNeighbourhood                     ▹ Algorithm 29:     **end if**10:     UpdateDataStructures                     ▹ Algorithm 411:   **end for**12:   StartOfLine:=True13:**end for**


For the new scan order, it is convenient to define a precedence operator, ≺, based on the order in which pixels are encountered during processing. Given two pixels, $P_{1}$ and $P_{2}$, then
(1)$$P_{1}≺P_{2}=\left\{\begin{array}{cc} true & \mathrm{when}\;P_{1}.y<P_{2}.y\\ true & \mathrm{when}\;\left(P_{1}.y=P_{2}.y\right)\wedge \left(P_{1}.y\;\mathrm{mod}\;2=0\right)\wedge \left(P_{1}.x<P_{2}.x\right)\\ true & \mathrm{when}\;\left(P_{1}.y=P_{2}.y\right)\wedge \left(P_{1}.y\;\mathrm{mod}\;2=1\right)\wedge \left(P_{1}.x>P_{2}.x\right)\\ false & \mathrm{otherwise}. \end{array}\right.$$

Precedence is used to select which label is the representative label during merger operations, and to determine when a connected component is completed.

Three auxiliary data structures are required for connected components analysis:The row buffer, RB[], saves the provisional labels assigned in the current row for providing the neighbourhood when processing the next row. Although the row buffer needs to manage pixels processed in a zig-zag scanned order, it is indexed within the following algorithms by logical pixel position.The merger table, MT[], indexed by label. This is to provide the current representative label for a component, given a provisional label. However, as a result of chains, more than one lookup in MT may be required.The data table, DT[], also indexed by label. This is to accumulate the feature vector extracted from each component. IFV(X) is the initial feature vector to be accumulated from the current pixel, and ∘ is the binary operator which combines two feature vectors.

Additional variables and arrays will be defined as required in the following algorithms.

### 2.2. Update Neighbourhood

Since the input pixels are streamed, moving from one pixel position to the next involves shifting pixels along within the neighbourhood window. Algorithm 2 indicates how the neighbourhood is updated during normal processing. A merger can only occur between pixels *A* and *C*, or *D* and *C* [26], and if both *A* and *D* are object pixels then they will already have the same label (from processing the previous window position). Therefore, the neighbourhood can be optimised with $L_{A\;or\;D}$ being the label $L_{A}$ or $L_{D}$ as required. The use of a superscript −, as in $L_{p}^{-}$, indicates the label $L_{p}$ at the end of the previous iteration.


**Algorithm 2**
UpdateNeighbourhood

1:**if**$L_{B}^{-}$**then**         ▹ Select $L_{A\;or\;D}$ based on whether *A* (previous *B*) is an object pixel2:   $L_{A\;or\;D}:=L_{B}^{-}$      ▹ Next value of $L_{A}$3:
**else**
4:   $L_{A\;or\;D}:=L_{X}^{-}$      ▹ Next value of $L_{D}$5:
**end if**
6:
$L_{B}:=L_{C}^{-}$
7:$L_{RB}:=RB\left[C\right]$       ▹ Look up position *C* in the row buffer8:**if**$L_{RB}$**then**         ▹ An object pixel is coming into neighbourhood9:   **if**
$\neg L_{C}^{-}$
**then**        ▹ It is the first object pixel after a background pixel10:     $L_{MT}:=MT\left[L_{RB}\right]$  ▹ First lookup in merger table11:     **if**
$L_{MT}=L_{RB}$
**then**  ▹ Label was representative label12:        $L_{C}:=L_{MT}$13:     **else**14:        $L_{C}:=MT\left[L_{MT}\right]$  ▹ Second lookup in merger table to get representative label15:        **if**
$L_{C}\neq L_{MT}$
**then**  ▹ Label change on second lookup indicates a chain16:          $MT\left[L_{RB}\right]:=L_{C}$   ▹ Update merger table to unlink the chain17:        **end if**18:     **end if**19:   **else**           ▹ Part of a run of consecutive pixels20:     $L_{C}:=L_{C}^{-}$        ▹ Repeat latest label21:     **if**
$L_{RB}\neq L_{RB}^{-}$
**then**   ▹ Label has changed, indicating a chain of mergers22:        $MT\left[L_{RB}\right]:=L_{C}$  ▹ Update merger table to unlink the chain23:     **end if**24:   **end if**25:
**else**
26:   $L_{C}:=0$        ▹ Lookup of background is unnecessary27:
**end if**



As the neighbourhood window pixels are shifted along, the new value for position *C* is obtained from the row buffer (line 7). If this is a background pixel, it is simply assigned label 0 (line 26). Note that if *C* is outside the image, for example when processing row 0 or when *X* is the last pixel in processing a row, then the background label (0) is used.

The row buffer provides the provisional labels assigned when processing the previous row. Although this label was the representative label for the component when it was written into the row buffer, subsequent mergers may mean that the label read from the row buffer is no longer the current representative label. It is necessary to look up the label in the merger table to obtain the current label (line 10). In a run of consecutive object pixels, all will belong to the same object, and will have the same label. The last label assigned to the run in the previous row will be the first read from the row buffer (as a result of the zig-zag scan), so only this label (see line 9) needs to be looked up in MT.

As a result of chains of mergers, a single lookup is not sufficient in the general case. Provided that the merger table is updated appropriately, two lookups may be required to give the current representative label. If the first lookup returns the same label (line 11), then that label has been unchanged (and is the representative label). However, if the first lookup returns a different label, then the provisional label may be stale and a second lookup is necessary (line 14). If the second lookup does not change the label, then this indicates that the single lookup was sufficient. If the second lookup returns a label that is different again, then this is part of a chain, and the value returned will be the current representative label.

To avoid having to lookup more than twice, it is necessary to update the merger table so that subsequent lookups of the original label produce the correct representative (line 16). This merger table update compresses the path, and performs the unchaining on-the-fly.

Within a run of consecutive object pixels, the representative label does not change. The latest label (after any merger at the previous window location, see line 20) is simply reused for *C*. If the row buffer output changes within a run of consecutive object pixels, this indicates that a merger occurred when processing the previous row and the provisional label from RB is out-of-date. This chain is unlinked, compressing the path by updating MT for the new label (line 22).

At the end of each row, it is necessary to reinitialise the window for the next row. As the window moves down, the pixels in the current row become pixels in the previous row. It is also necessary to flip the window to reflect the reversal of the scan direction. Algorithm 3 gives the steps required. Note that this is in place of Algorithm 2 for the first pixel of the next row.


**Algorithm 3**
ReverseNeighbourhood

1:$L_{A\;or\;D}:=0$  ▹This is now off the edge of the image2:$L_{B}:=L_{X}^{-}$   ▹Moving down makes current row into previous row3:
$L_{C}:=L_{D}^{-}$



### 2.3. Update Data Structures

Updating the data structures involves the following: assigning a provisional label to the incoming pixel based on the neighbourhood context; updating the merger table when a new label is assigned, or when a merger occurs; updating the feature vectors within the data table, and detecting when a connected component is completed. These are detailed in Algorithm 4.

A merger can only occur when *B* is a background pixel and $L_{A\;or\;D}$ is different from $L_{C}$ [26]. This condition corresponds to the block beginning line 3. The earliest assigned of $L_{A\;or\;D}$ or $L_{C}$ is selected as the representative label, and the other label is no longer used. The feature vectors associated with the two labels are merged, with the feature vector of the current pixel merged with the combination.

A new label is assigned to $L_{x}$ when $L_{A\;or\;D}$, $L_{B}$ and $L_{C}$ are background (line 15). New labels are assigned from the labelling recycling first-in-first-out (FIFO) buffer. Consequently, the label numbers are not in numerical sequence, so to determine precedence under merger conditions it is necessary to augment the labels with the row number (line 17). The feature vector for the new component is initialised with the feature vector of the current pixel, IFV(X).

If there is exactly one label in $L_{A\;or\;D}$, $L_{B}$ or $L_{C}$, it is assigned to $L_{X}$ and its feature vector in the data table at $DT[L_{X}]$ is merged with the feature vector of the current pixel IFV(X), as shown in lines 26 and 30.

A connected component is finished when it is not extended into the current image row. To detect this, an active tag, AT, field is introduced within the data table, DT. For each label, AT stores the 2D coordinates on the following image row beyond which no further pixels could be added to the component. When the scan passes this point on the following row (line 34), it is determined that the component is completed, enabling the feature vector to be output and the label recycled. The initial feature vector for the active tag is
(2)$$IFV\left(X\right).AT=\left\{\begin{array}{cc} \left(y+1,x-1\right) & \mathrm{when}\;y\;\mathrm{is}\;\mathrm{even},\\ \left(y+1,x+1\right) & \mathrm{when}\;y\;\mathrm{is}\;\mathrm{odd}. \end{array}\right.$$


**Algorithm 4**
UpdateDataStructures

1:**if**I[X]**then**                        ▹Object pixel2:   **if**
$\neg L_{B}$
**then**3:     **if**
$L_{A\;or\;D}\wedge L_{C}\wedge L_{A\;or\;D}\neq L_{C}$
**then**            ▹ Merger operation4:        **if**
$L_{A\;or\;D}.rw\leq L_{C}.rw$
**then**               ▹ Propagating merger5:          $L_{X}:=L_{A\;or\;D}$                     ▹ Assign representative label6:          $L_{old}:=L_{C}$7:          $L_{C}:=L_{X}$                     ▹ Update neighbourhood label8:        **else**9:          $L_{X}:=L_{C}$                     ▹ Assign representative label10:          $L_{old}:=L_{A\;or\;D}$11:        **end if**12:        $MT\left[L_{old}\right]:=L_{X}$                  ▹ Record merger in table13:        $DT\left[L_{X}\right]:=DT\left[L_{X}\right]\circ DT\left[L_{old}\right]\circ IFV\left(X\right)$     ▹ Merge data (and active tags)14:        $L_{old}\to LabelFIFO$                ▹ Recycle the old label15:     **else if**
$\neg L_{A\;or\;D}\wedge \neg L_{C}$
**then**              ▹ New label operation16:        $L_{X}:=newLabel$ (←LabelFIFO)         ▹ From a recycle queue17:        $L_{X}.rw:=y$                     ▹ Augment label with row number18:        $MT\left[L_{X}\right]:=L_{X}$                  ▹ Initialise merger table19:        $DT\left[L_{X}\right]:=IFV\left(X\right)$               ▹ Start feature vector20:     **else**21:        **if**
$L_{A\;or\;D}$
**then**                   ▹ Copy $L_{A\;or\;D}$22:          $L_{X}:=L_{A\;or\;D}$23:        **else**                       ▹ Copy $L_{C}$24:          $L_{X}:=L_{C}$25:        **end if**26:        $DT\left[L_{X}\right]:=DT\left[L_{X}\right]\circ IFV\left(X\right)$         ▹ Add current pixel to data table27:     **end if**28:   **else**                         ▹ Copy $L_{B}$29:     $L_{X}:=L_{B}$30:     $DT\left[L_{X}\right]:=DT\left[L_{X}\right]\circ IFV\left(X\right)$         ▹ Add current pixel to data table31:   **end if**32:
**else**
33:   $L_{X}:=0$                      ▹ Background pixel34:   **if**
$DT[L_{A}].AT=X$
**then**             ▹ Check completed object35:     **Output:**
$DT[L_{A}]$36:     $L_{A}\to LabelFIFO$               ▹ Recycle the label37:   **end if**38:
**end if**
39:$RB\left[X\right]:=L_{X}$                  ▹ Save label in row buffer for next row


For a label copy operation and a label merger operation, the active tag is updated along with the rest of the feature vector. The combination operator ∘ for two active tags is realised by applying precedence as defined in Equation (Equation 1) to select the later of the two active tags.
(3)$$AT_{1}\circ AT_{2}=\left\{\begin{array}{cc} AT_{2} & \mathrm{when}\;AT_{1}≺AT_{2},\\ AT_{1} & \mathrm{otherwise}. \end{array}\right.$$

For an efficient hardware implementation, it is sufficient to store only the least-significant bit of *y* for each active tag entry.

Figure 5 illustrates the update of active tags and detection of completed connected components. At the start of processing row 4, there 3 components with active tags as listed. Since row 4 is even (scanning left to right), the active tags are on the right hand end of the respective components. At (4,1), component 3 is extended and the active tag updated to (5,0)—the last possible scan position that could extend the current component 3. Similarly, at (4,5) component 2 is extended. At (4,6), components 1 and 2 merge with label 1 being retained as the representative label. Label 2 is recycled, and the active tags of labels 1 and 2 are combined. Further extensions of label 1 do not affect the active tag because the corresponding pixel active tags occur earlier in the scan sequence. When scanning back on row 5, label 1 is not extended, so when pixel (5,4) is a background pixel, the component labelled 1 is detected as completed, the feature vector output, and the label recycled. Similarly, at (5,0) component labelled 3 is detected as completed.

## 3. Architecture

Within this section, the hardware architecture to realise this algorithm is described. The input pixel stream is continuous, with one 1-bit binary pixel per clock cycle. Since there are no blanking periods, a streaming protocol based on AXI4-Stream [27] (advanced extensible interface) is used throughout the design. The modified protocol shown in Figure 6 has two control bits, one indicating the last pixel in every row, and one indicating the last pixel in every frame.

### 3.1. Zig-Zag Scan

The raster scanned input stream must be converted to a zig-zag ordered stream, where the odd numbered rows are presented in reverse order. Although this could easily be achieved with double buffering (reading the previous row from one buffer while writing the current row into a separate buffer) it can also be accomplished with a single row buffer with the access pattern shown in Figure 7.

After row 0 is initially written into the buffer, reading and writing are performed at the same address, with the raster based input stream being written into the same location that the zig-zag stream is read from. This requires switching the address sequence direction every second row. Converting the raster scan to a zig-zag scan introduces a latency of one row and one pixel.

The row buffer must also be modified to operate with a zig-zag scan pattern. Since successive rows are processed in the opposite order, the labels for each row must be read out in the reverse order that they were written. Data coming in for the new row overwrites the old data (already read out) in the buffer. As demonstrated in Figure 8, this can be accomplished by reversing the scan direction each row, effectively storing each label at the row buffer memory address corresponding to its *x* position.

### 3.2. Merger Table Processing

The label read from the row buffer may no longer be the current representative label as a result of mergers. For the look up operations performed in lines 7, 10, and 14 of Algorithm 2 it is necessary to look up the label in the merger table up to two times to obtain the current label. This is similar to the double lookup algorithm proposed in [6].

Although some labels may require two lookups, a single read port of a dual-port on-chip memory is sufficient for the merger table because it is unnecessary to look up every label from the row buffer. Labels of background pixels do not need to be looked up—all background pixels are simply labelled 0. In a sequence of consecutive object pixels, it is only necessary to look up the label of the first pixel in the sequence. An object pixel will either be followed by another object pixel or by a background pixel, neither of which need to be looked up, giving sufficient bandwidth for the two lookups.

Since each memory access requires 1 clock cycle (for synchronous memories such as the random access memory (RAM) blocks on most current FPGAs), it is necessary to pipeline the processing over 5 clock cycles as shown in Figure 9. The memory accesses are scheduled in advance so that the labels are available in the neighbourhood for assigning a label to the current pixel in stage 4.

As a result of pipelining, the write to the row buffer is delayed from the read by four clock cycles. This necessitates using a dual-port memory for the row buffer. The merger table is also dual-port, with the read port used for determining the representative label in stages 2 and 3 of the pipeline. The write port for the merger table is used for initialising the merger table when a new label is assigned (line 15), and for updating the merger table during merger operations (line 3). Both new label and merger operations occur in stage 5 of the pipeline. Unchaining of stale labels is also performed as the stale labels are encountered during the neighbourhood update (Algorithm 2) in stage 4 of the pipeline. The detailed architecture for implementing this is shown in Figure 10.

With synchronous memory, each read from an on-chip memory block is stored into a register; these are $L_{RB}$ and $L_{MT}$ for the row buffer and merger table respectively. The address for the merger table read comes either from $L_{RB}$ for the first read, or $L_{MT}$ for the second. Register $L_{MT2}$ is a pipeline register to hold the data if only a single read is required, with a multiplexer selecting the output of $L_{MT2}$ or $L_{MT}$ as the representative label. The conditional statements in Algorithm 2 are shown in blue in Figure 10, and are used to provide control signals for selecting appropriate multiplexer inputs.

In terms of forming the neighbourhood, $L_{C}$ is not directly registered, but is the output of multiplexers selecting the appropriate source register for $L_{C}$. $L_{B}$ and $L_{A\;or\;D}$ are registers. The current label output, $L_{X}$ is not registered, but is the output of the combinatorial logic which assigns a label to the current input pixel. This output is registered as $L_{D}$, available in the following clock cycle for window reversal at the end of each row, and for updating the merger table in pipeline stage 5 (if required). For row reversal, $L_{C}$ is assigned $L_{D}$ (Algorithm 3); however, since $L_{C}$ is not a register, it is necessary to insert a pipeline register, $L_{Crev}$.

Unchaining updates the merger table in pipeline stage 4. The data from line 16 is naturally available in that stage, but line 22 is detected at stage 2. It is necessary to delay both the address and data until stage 4. The address is delayed by pipeline registers $L_{RB1}$ and $L_{RB2}$, with the data coming from $L_{C}$, which at that stage in a run of consecutive pixels, is the feedback path from $L_{B}$ (line 20). For updating the merger table as a result of label assignment, for a new label, both the address and data come from $L_{D}$ (line 18). In the case of a merger, $L_{old}$ registers the old label, and is used for the address for the merger table update.

The dataflow for label assignment is shown in Figure 11. The binary input pixel is used to directly provide a control signal. The first multiplexer selects the label to propagate from the neighbourhood, with the second multiplexer selecting the background label (0), or a new label from the *LabelFIFO* (lines 16, 22, 29, 24 and 33). To reduce the logic requirements, the test for a background pixel on the row buffer output is simply pipelined through a series of registers to indicate whether $L_{C}$, $L_{B}$ or $L_{A\;or\;D}$ are object or background pixels.

### 3.3. Data Table

The final key section of the architecture is that which manipulates the data table. Figure 12 shows the data flow for the update and completed object detection. The inputs come from neighbourhood processing, after registering to pipeline the processing. The current pixel label, $L_{X}$ therefore, comes from the $L_{D}$ register, and $L_{old}$ in the case of mergers comes from the corresponding register in Figure 10. Data table processing is pipelined over three clock cycles, with the first cycle reading existing data from the data table when required, the second clock cycle is used to calculate the new feature vector, with the result being written to the data table (where necessary) in the third cycle. The neighbourhood position must also be registered twice before deriving the initial feature value (IFV) to maintain synchronisation. Control signals come from label assignment, whether it is a propagating label, a new label, a merger, or background pixel. Each of these cases will be described in turn.

For a propagating label, the neighbourhood had only a single label, which is copied to the current object pixel. If the previous pixel was a background pixel, then it is necessary to read the existing feature vector from the data table first. Otherwise, the feature vector will be available in the data table cache ($DT_{c}$) from processing the previous pixel. The initial feature vector, IFV, derived from the neighbourhood position is combined with the existing data, and the result stored in the data table cache, $DT_{c}$. The resulting feature vector is written back to the data table only when a background pixel is reached.

A new label operation has no existing data to load; the data table cache, $DT_{c}$, is simply initialised with the initial feature vector, IFV, in the second clock cycle.

A merger is a little more complex, because it may require two entries to be read from the data table. If the previous pixel was an object pixel, then the feature vector associated with $L_{A\;or\;D}$ will be available in $DT_{c}$. However, if the previous pixel was a background pixel, then data will not be cached for $L_{A\;or\;D}$. To overcome this problem, when the current pixel is a background pixel, $L_{B}$ is looked up in the data table. If $L_{B}$ is the label of an object pixel, then on the next clock cycle, it becomes $L_{A\;or\;D}$ and will be available in the cache. A merger will trigger the loading of $L_{C}$, so that it can be combined with $L_{A\;or\;D}$ and IFV. During the second clock cycle, $DT[L_{old}]$ is invalidated, enabling the label to be recycled. On the third clock cycle, the merged feature vector is written back to the data table.

Preloading the data table cache also facilitates detection of completed objects. From Algorithm 4 line 34, when the active tag (AT) of a completed object is the current pixel position, the last pixel will be in neighbourhood position *A*. At least the last three pixels (including the current pixel) will also have been background pixels otherwise they would have extended the object. Therefore, looking up $L_{B}$ when the current pixel is a background pixel gives the feature vector (containing AT) in the following clock cycle, enabling completed object detection (shown in blue in Figure 12). When the completed object is output, the data table entry is available for reuse by recycling the label.

## 4. Analysis

As a result of pipelining the computations, there are potentially data hazards, particularly in the use of memory for tables (the row buffer, merger table and data table), resulting from when data is expected to be in the table, but has not yet been written.

### 4.1. Row Buffer

For the row buffer, this can only occur at the end of the row, when the readout direction changes. The data hazards are demonstrated in Figure 13.

The last pixel of the previous row, **S**, is read from the row buffer when the neighbourhood window is at position **X** (as a result of pipelining). In the following clock cycles, reads from the row buffer begin their backward scan of the next row. However, pixel positions **T**, **U**, and **V** have not yet been written to the row buffer (or even assigned labels in the case of **T** and **U**). At the end of the row, lookup of positions **T** and **U** in the row buffer is actually unnecessary, because their values come directly from the neighbourhood when the window moves to the next row (Algorithm 3). Rather than read position **T**, it can simply be treated as a background pixel (label 0). This ensures that when the neighbourhood is at location **T**, neighbourhood position *C* (which is off the edge of the image) is correctly assigned a 0 (shaded pink in Figure 13). Similarly, position **U** is copied directly from the previous neighbourhood when the neighbourhood reverses direction. The row buffer output for **U**, too, can simply be treated as a background pixel. Finally, position **V** is read in the same clock cycle as it is written. This requires that the row buffer support a write-before-read semantic, or bypass logic be added to forward the value being written to the output.

### 4.2. Path Compression

Since both path compression and label assignment have write access to the merger table, it is necessary to check that these will not clash by attempting to write simultaneously. The possible scenarios are illustrated in Figure 14.

A new label and merger both update the merger table in pipeline stage 5. This is the clock cycle immediately following the label assignments, as illustrated in scenarios ① and ② respectively. Unchaining is performed in pipeline stage 4, corresponding to the clock cycle when the pixel appears in the neighbourhood window. This is illustrated in scenario ③ after two lookups, and scenarios ④ and ⑤ for a changed label within a consecutive run of pixels.

There cannot be a conflict between a change within a run, and a new label, because the change would require at least one pixel within the neighbourhood, preventing a new label assignment. Similarly, there can also be no conflict between a merger and a two lookup stale label because the merger would require $L_{C}$ to be non-zero, so the following pixel cannot be the first in a run. However, there can be conflicts between a new label and a two lookup stale label (scenario ⑥ in Figure 14b), and also between a merger and changed label in a run (scenario ⑦).

Where there is a conflict, the update resulting from the new label or merger should be deferred, with the stale label update taking priority. If the new label is followed by a merger (as in Figure 14b) then only the merger needs to be saved. This requires adding an additional storage register and multiplexer to the data path, and appropriate control logic. The maximum delay is two clock cycles, corresponding to ⑧, because changed labels in a run can occur at most every second pixel.

### 4.3. Merger Table

Potential data hazards can occur with the merger table, when data is read from the table before it is updated either as a result of merger or during the path compression.

A merger hazard is shown in Figure 15 for label 3. When scanning row 4, label 2 is read from RB[4], and is determined to be a representative label after a single lookup, MT[2]. Two clock cycles later, when the neighbourhood window is centred on pixel (4,3), component segments associated with labels 1 and 2 merge, with MT[2]←1 in the following cycle. Meanwhile, label 3 is read from RB[6], and requires two lookups in MT. The second lookup occurs in the same clock cycle that the merger is being written to MT, so the second lookup would actually return the old label (2), shown in red in Figure 15, and is not recognised as a stale label. A consequence of this is that pixel (4,6) would incorrectly be assigned label 2 rather than 1. To avoid this problem, the memory used for the merger table must also support the write-before-read semantic, or data forwarding be used to correctly return label (1) from the second lookup. Label 3 is then recognised as stale, and the merger table updated with MT[3]←1 as shown in green.

Delaying the merger table update after a merger (as described in the previous section) does not introduce any additional hazards because the run of pixels which induces the delay would also delay the start of the following run.

In a chain of successive mergers, such as in Figure 2, the previous merger is unlinked or compressed during the first merger table lookup, enabling the second lookup to provide the representative label. There are no data hazards associated with this process.

### 4.4. Data Table

Hazards within the data table can occur because the updated feature vector is written two clocks after the feature vector is read from the table. Alternating background and object pixels, with the object pixels belonging to the same connected component, can, therefore, cause a problem since the same label is being read from and written to in the same clock cycle. This can be solved if the memory supports read-before-write, or by adding bypass detection logic (the feedback data path from $DT_{c}$ to $DT_{i}$ is already present).

The other issue with the data table is detecting components which complete on the last pixel of a row, and on the row of the image. Equation (Equation 2) can be extended to include
(4)$$IFV\left(X\right).AT.y=\left\{\begin{array}{cc} H-1 & \mathrm{when}\;y=H-1,\\ y+1 & \mathrm{otherwise}; \end{array}\right.$$
(5)$$IFV\left(X\right).AT.x=\left\{\begin{array}{cc} 0 & \mathrm{when}\;x=0\;\mathrm{and}\;AT.y\;\mathrm{is}\;\mathrm{even},\\ x-1 & \mathrm{when}\;x\neq 0\;\mathrm{and}\;AT.y\;\mathrm{is}\;\mathrm{even},\\ W-1 & \mathrm{when}\;x=W-1\;\mathrm{and}\;AT.y\;\mathrm{is}\;\mathrm{odd},\\ x+1 & \mathrm{when}\;x\neq W-1\;\mathrm{and}\;AT.y\;\mathrm{is}\;\mathrm{odd}. \end{array}\right.$$

Thus, an object on the last line will be detected as complete in the clock cycle following the last pixel for that object.

## 5. Comparison and Discussion

In this section the proposed CCA algorithm is analysed with regards to throughput, latency and required hardware resources, and compared to other state-of-the-art CCA algorithms. For the comparison we chose the most recent CCA algorithms that are targeted for a realisation as hardware architectures [6,18,19,20,21,23,25].

### 5.1. Memory Requirements

The on-chip memory size and scalability with increasing image size was identified to be one of the most important criteria for a CCA hardware architecture to achieve a high-throughput for high-resolution image streams [6,18]. Therefore, the scalability of the on-chip memory is further examined in the following. As the algorithm by Jeong et al. [23] uses registers to realise the row buffer, both registers used as memory and on-chip memory (RAM blocks) are considered in the comparison of memory resources.

Table 1 compares the on-chip memory and register requirements for the algorithms presented in [6,20,21,23,25] for an image of size W×H. The number of labels required, $N_{L}$, defines the number of connected components that are stored at any one time inside an architecture before their feature vectors are ultimately output. $N_{L}$ is, therefore, the key factor for all architectures, as it defines the lower bound of the depth and the width for the memories of the examined CCA architectures. In their original publications the architectures extract different feature vectors. To enable a fair comparison, in Table 1 the width of a feature vector $W_{FV}$ containing the bounding box and the area is used for comparing the required memory. Table 2 shows the number of memory bits required for each data structure of the compared CCA architectures. The total numbers of on-chip memory and register bits are shown in Figure 16.

The architecture by Ma et al. [20] was the first to introduce relabelling to reduce the number of labels that are required, $N_{L}$, from $\frac{W\times H}{4}$ (in [18]) to $\frac{W}{2}$. The aggressive relabelling, however, requires two merger tables and two data tables to manage the labels changing from one row to the next. As shown in Figure 16 the architecture from [20], therefore, has the largest memory footprint among the compared CCA architectures.

The architectures of Klaiber et al. [6,21] use label recycling to improve memory-efficiency and, therefore, also require a maximum of $N_{L}$≈$\frac{W}{2}$ labels. Label recycling only requires a single data table and merger table, halving their size in comparison to [20] (although the augmented labels make the merger table wider). Since the on-chip memory requirements are dominated by the data table, this results in significant savings.

The architecture described in Jeong et al. [23] would scale with the image area, i.e., a maximum of $N_{L}=\frac{W\times H}{4}$ would be required for a worst case image. However, if feature vectors are output before the end of the image is reached, then those labels could be reused. Such label recycling is possible for the architecture in [23], even though it is not described (only merged labels are recycled). For a fair comparison, it is, therefore, assumed that the architecture scales with the image width, i.e., $N_{L}=\frac{W}{2}$ and the usage of an active tag (from [21]) for label recycling is assumed, even though it is not explicitly mentioned in the original publication. Directly replacing all instances of the old label within the row buffer enables many of the auxiliary data structures to be removed. Consequently, the modified architecture from [23] requires 30% less memory than [21]. This reduction, however, is only achieved because the row buffer is designed as context-addressable memory, which has to be realised with registers on FPGAs. The cyan-coloured bar in Figure 16 shows that almost one third of the required memory is realised directly by registers. For processing large image sizes, such as UHD8k, more than 90 kbits of registers are required to realise the row buffer and around 350 kbits of on-chip memory for the other data structures. Since modern FPGAs have a register to on-chip memory ratio between 1/20 and 1/60, a significant fraction of register resources are required. Furthermore, the routing effort on an FPGA, as well as the logic for addressing a content-addressable memory as large as 90 kbits consisting of registers is significant. An analysis of the scalability of such a context-addressable memory with increasing image size is not given in [23]. It seems unlikely that a context-addressable memory scales well on FPGAs, both, with maximum frequency and area. The number of registers required by the architecture of [23] is therefore a clear disadvantage when optimising for throughput or when minimising the FPGA resources.

The architecture of Tang et al. [25] represents a significant improvement, eliminating the need for the content addressable memory of [23] with only approximately 2% additional resources. The main reductions relative to [21] (approximately 30%) come from not needing to save the labels in the row buffer, and replacing the merger table with a linked list structure. Uniquely labelling each run also automatically recycles labels, eliminating the need for the recycle FIFO and active tag. For correct operation, however, it does require the first and last row and column of the image to be background. The results in Table 2 and Figure 16 do not include the logic required to extend the image with background pixels.

The proposed CCA architecture is an advancement of [6]. Due to zig-zag scanning, an additional memory structure to reorder incoming pixels from raster-scan order to zig-zag order is required. Since zig-zag processing resolves chains on the fly, the stale label stack and chain stack are no longer required. This reduces the amount of memory required by 9% compared to the architecture presented in [21]. Compared to [23,25] approximately 20% more memory bits are required, primarily from the merger table and other auxiliary data structures. The active tag is larger than that of [6] to detect object completion at the earliest possible time; this matches the timing of [25]. The advantage over [23] is merger handling using on-chip memories, rather than a large multiplexed shift register, which is a more efficient use of resources. The advantage over [25] is the removal of the requirement for the outside row and column of pixels to be background. The proposed architecture is also able to immediately detect completed objects in the final row as they complete.

### 5.2. Implementation Results

Table 3 shows the results of the CCA architecture implemented using VHDL on an Intel Cyclone V 5SEMA5F31C6 (using Quartus 17.1) and a Xilinx Kintex 7 xc7k325-2L (using Vivado 2016.4). These tables show the number of lookup tables (LUTs/ALUTs), registers (FF) and on-chip memory bits (and memory blocks) each component of the CCA architecture requires for processing UHD8k images. The slightly higher memory requirements for the Cyclone V for the merger table and data table are a result of the synthesis tools rounding the memory depth up to the next power of 2.

The scalability of the proposed CCA architecture with increasing image size is explored in Figure 17. The number of required number of LUTs/ALUTs is shown in Figure 17a. On the Intel Cyclone V the number of ALUTs increases logarithmically with the image width. On the Xilinx Kintex 7 the number of LUTs increases from VGA to HD1080 image size to almost 1400 and then drops to around 800 LUTs for UHD4k and UHD8k image size. This is a direct result of the usage of LUTs as distributed RAM to realise small memories. On Kintex 7 FPGAs this is done to prevent using valuable on-chip memory resources from being used inefficiently for small memories that only utilise a small fraction of the 18 kBit minimum size. From UHD4k all the memories are realised with RAMs. The number of LUTs from UHD4k to UHD8k image size, therefore, increases only marginally.

Figure 17b shows a small logarithmic increase in the number of registers required with image width for both FPGAs. The Cyclone V uses slightly more registers than the Kintex 7 as a result of register duplication during the place and route stage. The required on-chip memory bits scale linearly with the image width, as shown in Figure 17c. The only exception that can be observed is that for the Kintex 7 the same amount of block memory is required for the HD720 and HD1080 image sizes. This remains constant as a result of the usage of distributed RAM as indicated in Figure 17a. The small increase for the Cyclone V for the HD720 image size is simply a result of the discrete nature of the RAM blocks.

The throughput of the architecture is proportional to the maximum clock frequency. Therefore, it determines how well the throughput of the architecture scales with increasing image width. As shown in Figure 17d the maximum frequency remains almost constant for both FPGAs. A maximum frequency around 180 MHz can be reach on the Kintex 7 for all examined image sizes. For the Intel Cyclone V, the maximum frequency is around 105 MHz for all image sizes.

### 5.3. Comparison of CCA Hardware Architecture

Table 4 compares the results reported by Johnston et al. [19], Ma et al. [20], Klaiber et al. [21], Jeong et al. [23], and Tang et al. [25], with the implementation results of the proposed architecture. The reported results differ in image size, extracted feature vectors, the FPGA technology used and the maximum number of labels that can be stored in the architecture. A direct comparison of the architectures from Table 4 is, therefore, not meaningful. The differences of the results of the proposed architecture are discussed for each examined architecture in the following.

Comparison to [19,20]: The proposed architecture is an advancement of these architectures. The number of memory bits was significantly reduced by the introduction of label recycling and omission of the chain stack. The required LUTs and registers are mostly used for control logic and are, therefore, similar for the proposed architecture when comparing to [19,20].

Comparison to [21]: In the proposed approach the chain stack and stale label stack are no longer required, however, the memory for storing active tag has increased compared to [21]. The required on-chip memory could, therefore, be reduced up to 10%, as shown in Table 2. There was a small increase in maximum frequency (35 MHz for 256×256 images and by 10 MHz for UHD8k images). This was achieved due to the simplified label assignment. The critical path was in [21] in the label assignment. For the proposed architecture it is now in the calculation of the active tag in the data table. For the 256×256 image size, the required on-chip memory has decreased significantly from 108 kbits to 18 kbits. In [21] most data structures on the Kintex 7 occupy full 18 kbit RAM blocks even if a significant part is unused. The proposed architecture makes use of distributed RAM for small data structures; these are realised with LUTs. This also explains why the number of required LUTs has almost doubled from [21] to the proposed architecture for small image sizes. For the UHD8k image size, the LUT and register requirements are slightly higher than in [21] reflecting the more complex control, and the improved object completion detection. It should be noted that the results in Table 4 for [21] are for extracting the bounding box only, whereas the results for the proposed architecture are for extracting bounding box and area (which requires a wider data table).

Comparison to [23]: The relatively low RAM requirement of [23] is directly a result of restricting the design to 127 labels; this would grow significantly if the design increased $N_{L}$ to handle any image (the data table size is proportional to $N_{L}$). The number of registers is not directly reported in [23]. However, as the number of registers required for the row buffer is proportional to the image width (here 1920 for an HD image) and the label width (here 7 bits for 127 labels) it cannot be lower than 13,440 registers. As discussed in the analytical comparison from Table 1 and Figure 16, implementation of [23] requires significantly more registers than the other architectures while being limited to only 127 labels. The use of multiplexed registers for the row buffer would impact on the routability of the design, and this is the likely cause of the significantly lower clock frequency. The major advantage of the proposed architecture over [23] is that all of the data structures are realised as on-chip memories. This allows the proposed design to use a smaller FPGA device, as the number of registers required is much smaller and the proportion of on-chip memory and registers is closer to modern FPGAs.

Comparison to [25]: The small resource requirement comes from the simplified logic for maintaining the linked list data structures. Although the RAM requirements for the Virtex II seem anomalously large, the minimum RAM block size is 18 kbits, with the tools reporting the total size rather than just the number of bits used (the remainder of the RAM blocks are unusable). The RAM for the Cyclone IV is close to that indicated by Table 1. Again it should be noted that the results reported for Tang et al. are for extracting the bounding box only. Extracting the area as well requires a 50% wider data table, and would also require a small increase in the resources required. That said, the proposed architecture requires more resources and operates at a similar speed to [25]. It should be remembered, however, that Tang et al. requires the borders of the image to be background pixels. The logic reported does not include that required to either ensure this, or to pad the image if required.

### 5.4. Throughput

To compare the throughput of the architectures from Johnston and Bailey [19], Klaiber et al. [6,21], Ma et al. [20], Jeong et al. [23], Tang et al. [25] and the proposed architecture, the maximum number of clock cycles to process an image of size W×H is examined. All of the designs are capable of processing one pixel per clock cycle of the input image. The difference is the end of row processing for resolving chains, which are data-dependent.

For [6,19,21], the pattern which creates the maximum number of chain stack entries in an image is the stair pattern shown in Figure 18a. It adds an overhead of $\frac{W}{5}$ cycles to each image row to process the content stored in the chain stack and to update the merger table.

The architecture of [20] has a translation table directly connected to the output of the merger table, with many mergers managed by the translation of labels from one row to the next. This makes the pattern that creates the maximum number of chains more complicated, i.e., it repeats with a lower frequency than the pattern from [19,21]. In Figure 18b it is called the feather pattern. It adds an overhead of $\frac{W}{8}$ cycles to every second image row (giving an average of $\frac{W}{16}$ cycles per row).

The proposed architecture and the architectures of [23,25] are data-independent and do not have a chain stack. Therefore, they only require one clock cycle to process a pixel, with no end of row overhead for resolving chains. However, to process the complete image, [25] requires extending the image by 1 row and column on each side (i.e., to process the full image, the end of row overheads have not been completely eliminated). These results are summarised in Table 5.

Throughput also depends on the clock frequency. For each architecture and platform, the lowest clock frequency from Table 4 is selected, and scaled according to the overhead. From this, it is clearly seen in Table 5 that the proposed approach is 2 or 3 times faster than [23], primarily as a result of using memory for the row buffer rather than distributed registers. The reduction in overhead amplifies the small improvement in clock frequency over [6,21], giving a 26% improvement in throughput.

### 5.5. Latency

In terms of CCA, latency can be defined as the number of clock cycles from the time when the last pixel of a connected component is received until its feature vector is output by the CCA architecture. There is a small latency (of a few clock cycles) resulting from pipelining, but the majority comes from detecting component completion, which is dependent on the image width, *W*. Since the width term dominates, the small pipeline latency (which is constant) will be ignored in this discussion.

The architecture of Ma et al. [20] has two data tables, one for feature vectors of connected components of the previous row and one for the current row. If a connected component is extended from the previous row to the current row, its feature vector is moved from one data table to the other. A connected component that is finished is not extended to the current row, i.e., when the end of the current row is reached the associated feature vector is still in the data table for the previous row. While processing the next row this data table scanned to detect completed components and output the feature vector. Due to aggressive relabelling, connected components are stored in the order that they appear in the current image row. Therefore, an object at the start of the row will have a latency of 2W cycles, while those at the end of the row will have a latency of *W* plus a scan time within the data table of up to $\frac{W}{2}$ (depending on the number of separate components on the row) to detect the completed object.

In the architecture of Klaiber et al. [6,21], the data table is scanned for completed objects at the start of the second row after the last pixel of the object. The latency before this scan, therefore, ranges from *W* to 2W, depending on the position along the row. As a result of label recycling, the label could be anywhere within the data table, with the latency of detecting the completed component during the scan varying up to $\frac{W}{2}$. These combine to give an average latency of 1.75W up to a maximum of 2.5W.

The mechanism of Tang et al. [25] detects completed objects when it encounters a hanging label, i.e., the end of a list of runs on the previous row with no connection to the current row. This is the earliest time that a component can be detected as completed, and has a latency of *W* clock cycles. Note that the preprocessing to convert from 4-connectivity to 8-connectivity does not introduce any significant latency. However, padding the image to ensure that the image borders are background pixels will introduce an additional row of latency (*W* clock cycles—not reported here).

In the proposed design, converting from a raster scan to a zig-zag scan introduces an additional latency relative to the other methods. Therefore, to minimise latency, it is essential to detect completed components at the earliest possible opportunity (on the following row), which is achieved by the new completion detection mechanism. The latency of the zig-zag conversion is *W* clock cycles on even numbered rows, and between 0 and 2W clock cycles on odd numbered rows (during the reverse scan). The latency of detection is between 2W at the start of a scan of a row (to scan all of the row, and back again on the next row), through to 0 at the end of a scan. These combine to give a latency of between *W* and 3W, with an average latency of 2W clock cycles. If the zig-zag conversion is unnecessary (for example if streaming from memory in zig-zag order), then objects are detected as completed with a latency of between 0 and 2W, with an average of *W* clock cycles.

The algorithm of Johnston and Bailey [18,19] does not allow completed objects to be detected before the end of the image. Similarly, Jeong et al. [23] gives no criterion for detecting a finished connected component before the end of the image. The latency is, therefore, the number of cycles from the last pixel of a connected component until the end of the image. These architectures were, therefore, not compared in terms of latency. In principle, however, although not part of the architecture of [23], there is no limitation (apart from a few more resources) against detecting and outputting the feature vector in a manner similar to that used in [21], or indeed that proposed in this paper.

Table 6 summarises the latency of the architectures considered. Although the proposed architecture introduces significant latency in the conversion of the input to a zig-zag scan, this has been mitigated by the proposed new approach to completed object detection. The slight increase in latency is the price to pay for the increase in throughput from the elimination of end of row overheads. Note that the feature vectors of any objects touching the last row of the image will be output with almost no latency (only the pipeline delay), which is significantly shorter than any of the other architectures.

## 6. Summary and Conclusions

Pixel based hardware CCA architectures are designed to process streamed images at one pixel per clock cycle. However, with synchronous memories within modern FPGAs, this limits the designs to one memory access per clock cycle, which can create issues with stale labels resulting from chains of mergers. Current approaches manage this by resolving stale labels at the end of each image row, although this introduces a variable, image dependent, delay.

Jeong et al. [23] solved this by replacing the memory with a multiplexed shift register, enabling all instances of old labels to be replaced immediately. However, the movement away from a memory structure comes at a cost of considerably increased logic resources and registers and a lower maximum clock frequency.

Tang et al. [25] took a different approach, and rather than relabel the pixels which have already been seen, manages merger resolution through manipulation of pointers within a linked list structure. This eliminates the overheads associated with chains, and provides an efficient mechanism for detecting completed components and recycling labels. Although it claims to have no overheads, it does require the border pixels within the image to be background. This would require padding the image before processing, and results in two clock cycles overhead for each row.

In this paper, we have demonstrated an alternative approach to resolve stale labels on-the-fly by using a zig-zag scan. This allows continuous streamed images to be processed with no data dependent overheads, while retaining the use of memory for buffering the previous row.

The cost of this approach is slightly increased control logic over prior memory based approaches. This is to handle the zig-zag scan, and to manage multiple lookups within the merger table. The memory requirements are reduced because fewer auxiliary data structures are required. The presented design also allows a slightly higher clock frequency than prior state-of-the-art designs, in addition to the improved throughput. The use of memory rather than a multiplexed shift register makes it significantly faster than the architecture of [23].

Conversion from a raster scan to a zig-zag scan does increase the latency (in terms of the number of clock cycles). This has been mitigated to some extent by a new algorithm that detects when objects are completed at the earliest possible time. Overall, the proposed changes give an improvement over current state-of-the-art methods. 

## Figures and Tables

**Figure 1 jimaging-05-00045-f001:**
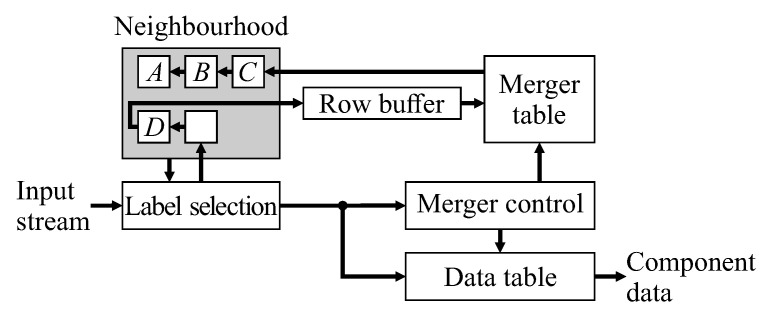
Basic architecture of single-pass connected components analysis.

**Figure 2 jimaging-05-00045-f002:**
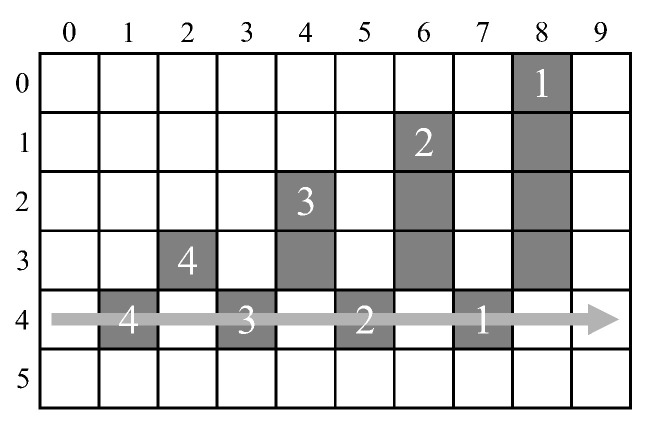
A chain of successive mergers: 4⇒3; 3⇒2; 2⇒1.

**Figure 3 jimaging-05-00045-f003:**
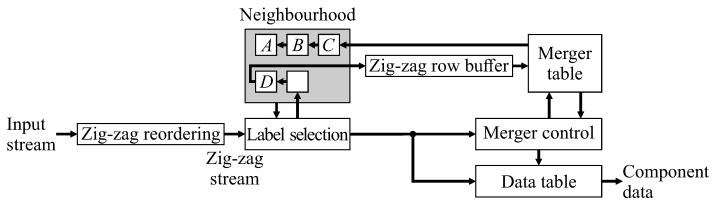
Basic architecture of zig-zag based single-pass connected components analysis.

**Figure 4 jimaging-05-00045-f004:**
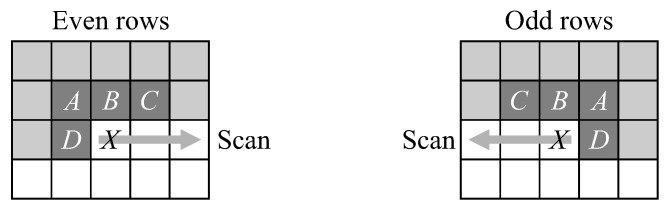
The neighbourhood of the current pixel, *X*, shaded dark. Shaded pixels have already been processed. Labelling is dependent on the scan direction.

**Figure 5 jimaging-05-00045-f005:**
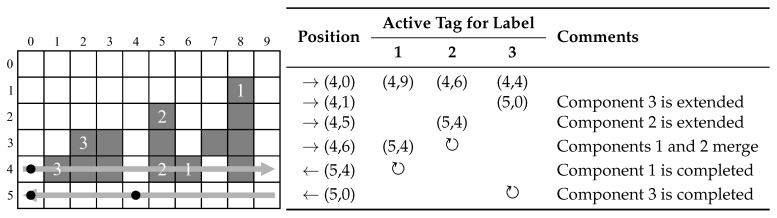
Example for detection of finished connected component at position *X*. ↻ indicates when the label is recycled.

**Figure 6 jimaging-05-00045-f006:**
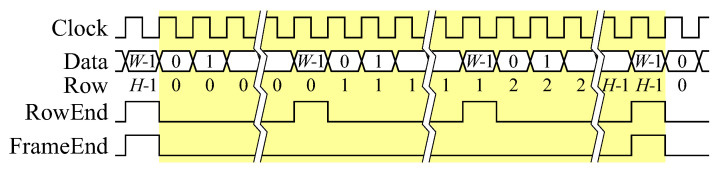
Continuous pixel stream protocol, with one image frame highlighted.

**Figure 7 jimaging-05-00045-f007:**
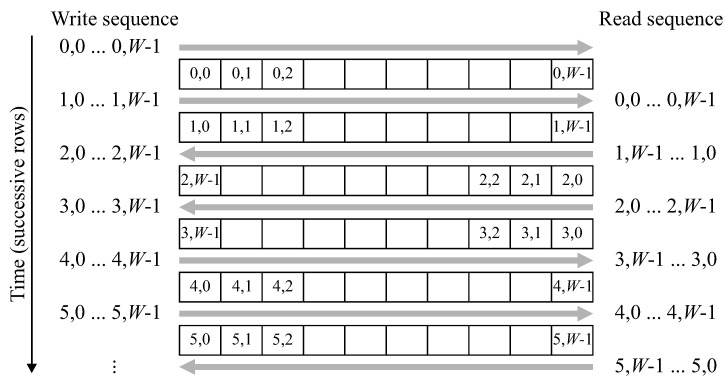
Operation of the zig-zag reordering buffer. Positions in the figure are shown in the format y,x, where *y* refers to the row and *x* to the column the pixel was assigned.

**Figure 8 jimaging-05-00045-f008:**
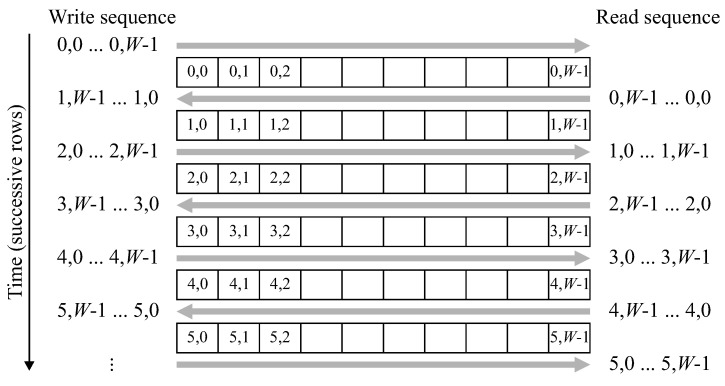
Operation of the row buffer with zig-zag ordered data.

**Figure 9 jimaging-05-00045-f009:**
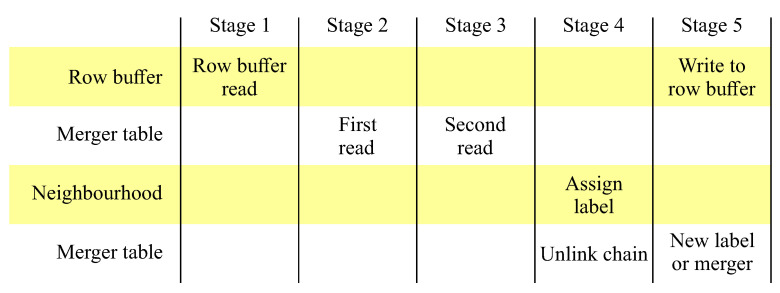
The 5 pipeline stages for processing each input pixel.

**Figure 10 jimaging-05-00045-f010:**
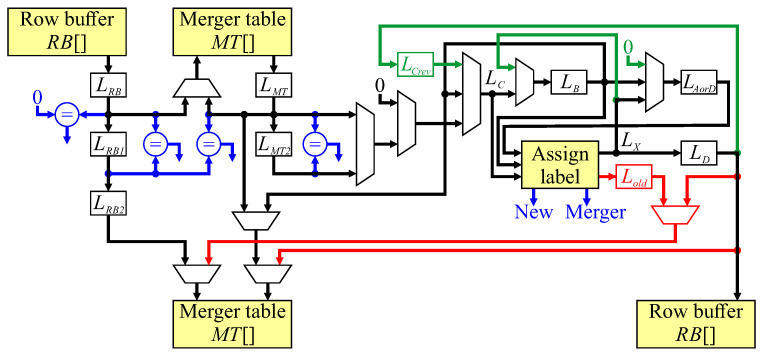
The detailed pipeline architecture for zig-zag connected components. Blue represents control signal generation, green indicates processing for end of row reversal, and red are the merger table updates for new label assignment and merger processing.

**Figure 11 jimaging-05-00045-f011:**
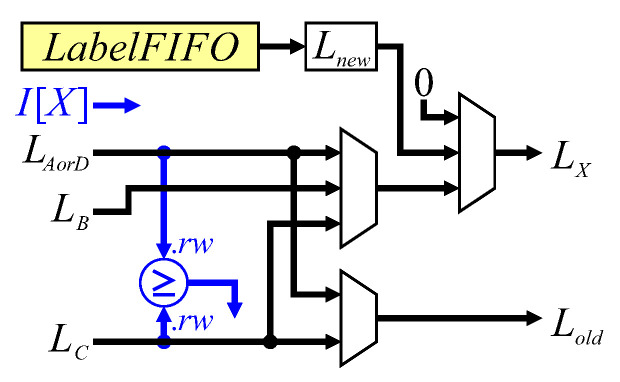
Architecture for label assignment. Blue represents control signal generation.

**Figure 12 jimaging-05-00045-f012:**
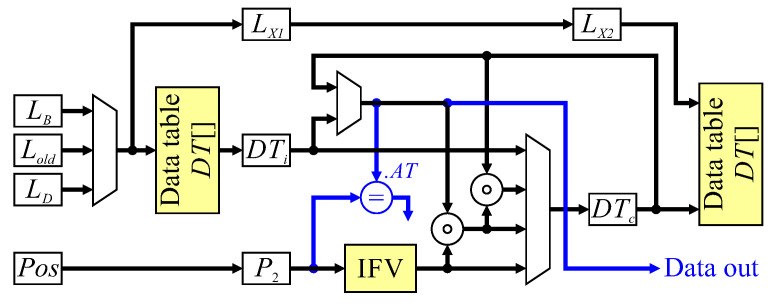
Architecture for data table update. Blue signals relate to detecting completed components.

**Figure 13 jimaging-05-00045-f013:**
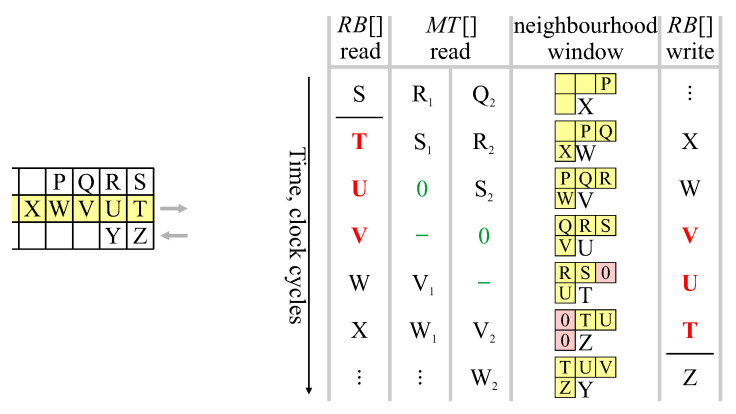
End of row timing, showing data hazards in red. Subscripts 1 and 2 refer to the first and second reads from the merger table (if required).

**Figure 14 jimaging-05-00045-f014:**
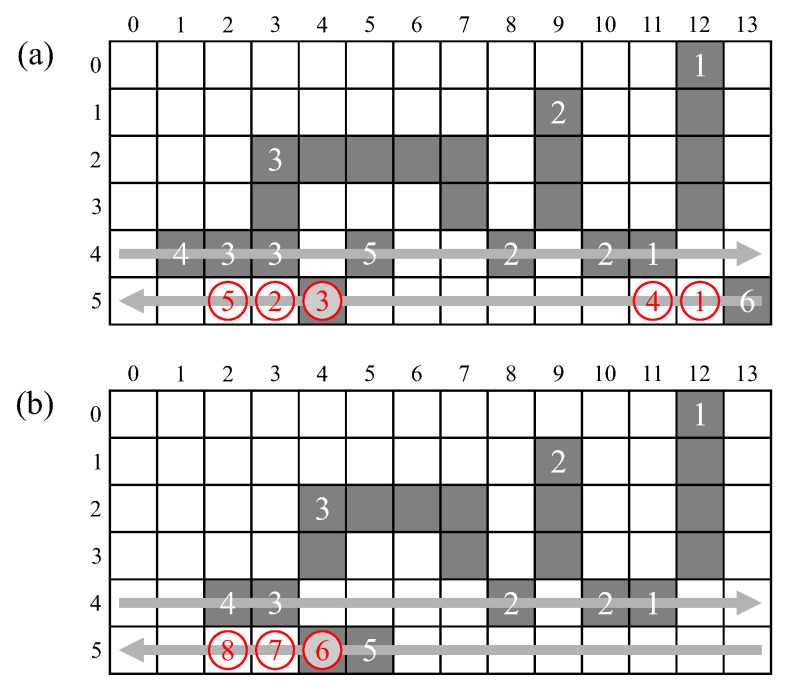
Accesses to update the merger table: (**a**) Scenarios with no conflicts; (**b**) Scenarios with conflicts.

**Figure 15 jimaging-05-00045-f015:**
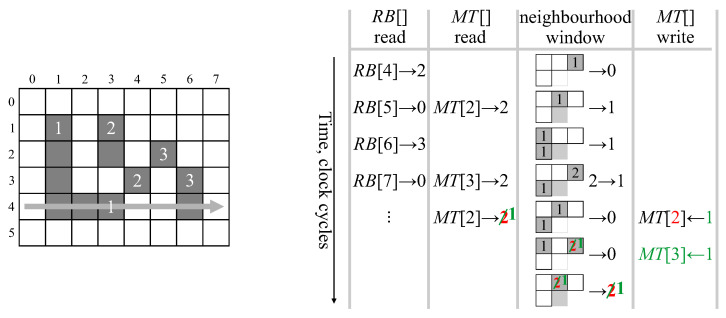
Timing of hazards associated with the merger table.

**Figure 16 jimaging-05-00045-f016:**
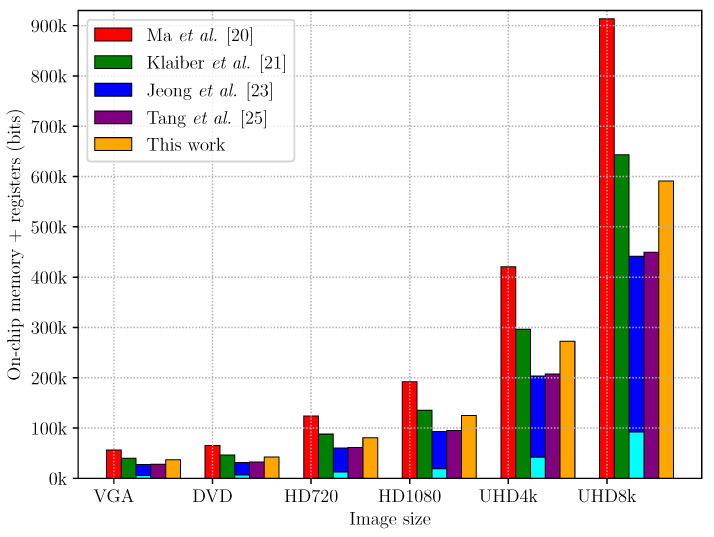
The bar diagram shows the number of on-memory and register bits that are required to process images of different sizes. The bars indicate on-chip memory. The cyan coloured part of [23] indicates the registers required for the row buffer.

**Figure 17 jimaging-05-00045-f017:**
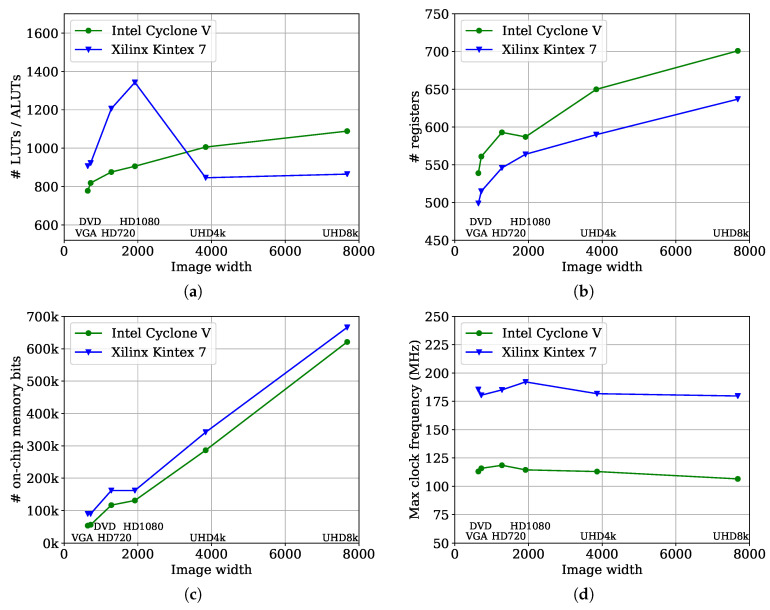
These diagrams that show the number of (**a**) look up tables (LUTs / ALUTs), (**b**) registers and and (**c**) on-chip memory bits for different image sizes for the implementation of the proposed CCA architecture on an Intel Cyclone V 5SEMA5F31C6 and a Xilinx Kintex 7 xc7k325-2L FPGA. In (**d**) the maximum clock frequency is shown.

**Figure 18 jimaging-05-00045-f018:**
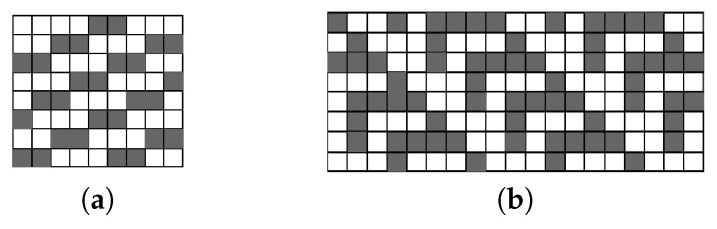
Image patterns that create the worst case average overhead for (**a**) [6,19,21] and for (**b**) [20].

**Table 1 jimaging-05-00045-t001:** Comparison of on-chip memory and register requirements. For all compared architectures the feature vectors are composed of bounding box and area for each connected component, i.e., the width of the feature vector, $W_{FV}$, is equivalent for all architectures, $W_{FV}=2\left\lceil \log_{2}W\right\rceil +2\left\lceil \log_{2}H\right\rceil +\left\lceil \log_{2}WH\right\rceil$.

	Ma et al. [20]	Klaiber et al. [6,21]	Jeong et al. [23]		Tang et al. [25]	This Work
Number of labels, $N_{L}$	$\left\lceil \frac{W}{2}\right\rceil$	$\left\lceil \frac{W+5}{2}\right\rceil$	$\left\lceil \frac{W}{2}\right\rceil$ to $\left\lceil \frac{W\;\times \;H}{4}\right\rceil$		$\left\lceil \frac{W}{2}\right\rceil$	$\left\lceil \frac{W}{2}\right\rceil$
Chain stack size, $N_{CS}$	$\left\lfloor \frac{W-1}{2}\right\rfloor$	$\left\lfloor \frac{W-1}{2}\right\rfloor$	−		−	−
Label width, $W_{L}$	$\left\lceil \log_{2}N_{L}\right\rceil$	$\left\lceil \log_{2}N_{L}\right\rceil$	$\left\lceil \log_{2}N_{L}\right\rceil$		$\left\lceil \log_{2}N_{L}\right\rceil$	$\left\lceil \log_{2}N_{L}\right\rceil$
Augmented label, $W_{AL}$	−	$W_{L}\;+\;\left\lceil \log_{2}H\right\rceil$	−		−	$W_{L}+\left\lceil \log_{2}H\right\rceil$
**Hardware Data Structure**	**RAM**	**RAM**	**Registers**	**RAM**	**RAM**	**RAM**
Zig-zag buffer, ZZ	−	−	−	−	−	W×1
Recyle FIFO, *R*	−	$N_{L}\;\times \;W_{L}$	−	$N_{L}\;\times \;W_{L}$	−	$N_{L}\;\times \;W_{L}$
Row buffer, RB	$W\;\times \;W_{L}$	$W\;\times \;W_{L}$	$W\;\times \;W_{L}$	−	W×2	$W\;\times \;W_{L}$
Merger table, MT	$2N_{L}\;\times \;W_{L}$	$N_{L}\;\times \;W_{AL}$	−	−	−	$N_{L}\;\times \;W_{AL}$
Chain stack, CS	$N_{CS}\;\times \;2W_{L}$	$N_{CS}\;\times \;2W_{L}$	−	−	−	−
Translation table, TT	$N_{L}\;\times \;W_{L}$	−	−	−	−	−
isRoot flag, *F*	−	$N_{L}\;\times \;1$	−	−	−	−
Active tag, AT	−	$N_{L}\;\times \;2$	−	−	−	$N_{L}\;\times \;\left(\left\lceil \log_{2}W\right\rceil \;+\;1\right)$
Stale label stack, SLS	−	$\left\lceil \frac{W}{10}\right\rceil \;\times \;W_{L}$	−	−	−	−
Linked lists, LL	−	−	−	−	$3N_{L}\;\times \;W_{L}$	−
Data table, DT	$2N_{L}\;\times \;W_{FV}$	$N_{L}\;\times \;W_{FV}$	−	$N_{L}\;\times \;W_{FV}$	$N_{L}\;\times \;W_{FV}$	$N_{L}\;\times \;W_{FV}$

**Table 2 jimaging-05-00045-t002:** Comparison of memory requirements of all data structures of the examined CCA architectures for different image sizes from VGA to UHD8k.

	VGA 640 × 480	DVD 720 × 576	HD720 1280 × 720	HD1080 1920 × 1080	UHD4k 3840 × 2160	UHD8k 7680 × 4320
Ma et al. [20]						
RB	5760	6480	12,800	19,200	42,240	92,160
MT	5760	6480	12,800	19,200	42,240	92,160
CS	5742	6462	12,780	19,180	42,218	92,136
TT	2880	3240	6400	9600	21,120	46,080
DT	36,480	42,480	79,360	124,800	272,640	591,360
**Total**	**56,622**	**65,142**	**124,140**	**191,980**	**420,458**	**913,896**
Klaiber et al. [6,21]						
*R*	2907	3267	6430	9630	21,153	46,116
RB	5760	6480	12,800	19,200	42,240	92,160
MT	5814	6897	12,860	20,223	44,229	96,075
CS	5742	6462	12,780	19,180	42,218	92,136
*F*	323	363	643	963	1923	3843
AT	646	726	1286	1926	3846	7686
SLS	576	648	1280	1920	4224	9216
DT	18,411	21,417	39,866	62,595	136,533	295,911
**Total**	**40,179**	**46,260**	**87,945**	**135,637**	**296,366**	**643,143**
Jeong et al. [23]						
*R*	2880	3240	6400	9600	21,120	46,080
RB	5760	6480	12,800	19,200	42,240	92,160
AT	640	720	1280	1920	3840	7680
DT	18,240	21,240	39,680	62,400	136,320	295,680
**Total**	**27,520**	**31680**	**60,160**	**93,120**	**203,520**	**441,600**
Tang et al. [25]						
RB	1280	1440	2560	3840	7680	15,360
LL	8640	9720	19,200	28,800	63,360	138,240
DT	18,240	21,240	39,680	62,400	136,320	295,680
**Total**	**28,160**	**32,400**	**61,440**	**95,040**	**207,360**	**449,280**
This work						
ZZ	640	720	1280	1920	3840	7680
*R*	2880	3240	6400	9600	21,120	46,080
RB	5760	6480	12,800	19,200	42,240	92,160
MT	5760	6840	12,800	20,160	44,160	96,000
AT	3520	3960	7680	11,520	24,960	53,760
DT	18,240	21,240	39,680	62,400	136,320	295,680
**Total**	**36,800**	**42,480**	**80,640**	**124,800**	**272,640**	**591,360**

**Table 3 jimaging-05-00045-t003:** Synthesis results targeting a UHD8k image (7680 × 4320). ALUTs are Intel’s adaptive lookup tables; FFs are the number of flip-flops or registers; M10K are the number of Intel’s 10 kbit RAM blocks; BRAMs are the number of Xilinx’s 36 kbit block RAMs.

Module	Intel Cyclone V 5SEMA5F31C6				Xilinx Kintex 7 xc7k325-2L			
	ALUTs	FFs	RAM (bits)	M10K	LUTs	FFs	RAM (bits)	36k BRAMs
Zig-zag buffer	28	19	7680	1	46	19	7680	0.5
Label generator	51	31	46,080	6	14	26	46,080	1.5
Row buffer	49	21	92,160	12	163	30	92,160	3
Merger table	99	103	102,400	13	219	101	96,000	3
Neighbourhood	226	252	0	0	95	217	0	0
Data table	635	275	372,736	46	470	244	322,560	10.5
Total	1088	701	621,056	78	867 ${}^{a}$	637	564,480	18.5

${}^{a}$ LUTs shared between multiple components are counted in both.

**Table 4 jimaging-05-00045-t004:** Comparison of several CCA hardware architectures. Abbreviations for the extracted feature vector are: (A) area, (C) component count, (FOM) first-order moment, (BB) bounding box.

Implementation of Architecture	Technology	Image Size (pixels)	Extracted FV	LUTs	Registers	RAM (bits)	$f_{\max}$ (MHz)
Johnston and Bailey [19] ${}^{a}$	Spartan II	670×480	C	620	271	12 k	N/A
			A	758	299	20 k	N/A
Ma et al. [20]	Virtex II	640×480	A, C	1757	600	72 k	40.64
Klaiber et al [21]	Kintex 7	256×256	BB	493	296	108 k	185.59
		7680×4320		818	444	548 k	170.53
Jeong et al. [23] ${}^{b}$	Cyclone IV	640×480	BB, FOM	36,478	N/A	18k	60.58
		1920×1080		57,036	N/A	29 k	58.44
Tang et al. [25]	Virtex II	256×256	BB	543	187	72k	104.26
	Cyclone IV			489	303	7287	122.94
This work	Cyclone V	256×256	BB, A	682	479	22 k	122.56
		7680×4320		1088	701	621 k	106.52
	Kintex 7	256×256	BB, A	882	503	18 k	220.02
		7680×4320		867	637	564 k	180.47

${}^{a}$ Hardware resources are for a maximum of 255 labels [19]. ${}^{b}$ Hardware resources are for a maximum of 127 labels [23].

**Table 5 jimaging-05-00045-t005:** Comparison of processing cycles for a W×H image.

Architecture	Number of Cycles	$f_{\max}$ (MHz)	Throughput (Mpix/s)
Johnston and Bailey [19]	6/5×W×H	N/A	N/A
Klaiber et al. [6,21]	6/5×W×H	170.53	142.11
Ma et al. [20]	17/16×W×H	40.64	38.25
Jeong et al. [23]	W×H	58.44	58.44
Tang et al. [25]	(W+2)×(H+2)	104.26	102.65
This approach	W×H	106.52	106.52 (Cyclone V)
		180.47	180.47 (Kintex 7)

**Table 6 jimaging-05-00045-t006:** Latency (in clock cycles) for an image of width *W*.

Architecture	Average Latency	Maximum Latency
Ma et al. [20]	1.75W	2W
Klaiber et al. [6,21]	1.75W	2.5W
Tang et al. [25] (without padding)	*W*	*W*
This approach (with zig-zag conversion)	2W	3W
This approach (without zig-zag conversion)	*W*	2W

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
