# Peer review of "Zig-Zag Based Single-Pass Connected Components Analysis"

_2313-433X, 2019, doi:10.3390/jimaging5040045_

Round 1
Reviewer 1 Report
This article presents a new CCA algorithm and its FPGA implementation.
The paper is clear, well written and well referenced. As far as I know, the algorithm is a novelty. It is an interesting solution to replace the additionnal resolution of the merger chain.
The three main parts are :
- Algorithm presentation (without hardware implementation)
- Hardware implementation with details on the circuits dedicated to the specific algorithms
- A rigourous analysis including competitors implementations.
I only report few details to improve further this excellent article.
Typos :
Page 3/24 line 85 (to is repeated)
Page 4/24 in 2.1 the first paragraph is repeated twice (101-104 and 105-108).
Page 14/24 line 343 additional is repeated
Page 17/24 line 414 the usage ON an AT --> the usage OF an AT ?
Page 18/24 line 454 numebr --> number
page 21/24 line 524 the reduction of IS overhead amplifies the small … --> OF THIS …
page 22/24 line 551 "this scan is therefore varies from …" I'm not sure of the varies usage.
Presentation improvement suggestions :
- Algorithm 1 line 3 and 4 : on 1 line if possible.
- Improve the aligment of write and read sequence to enhance readability for both figures 7 and 8 page 9
- Improve the aligment of Cyclone V and Kintex 7 results to enhance readability for table5
Questions :
- in table 1 (page15/24) :
* Is NL the number of ACTIVE labels
* Is WL the MAXIMUM Label Width
Author Response
We would like to thank the reviewer for their kind comments. Thank you for pointing out the typos – we had noticed some of these, but not all. All have been corrected now.
Algorithm 1, lines 3 and 4 are now on a single line
We are not sure what you mean by the alignment of the read and write sequences in figures 7 and 8 – the alignment is correct. After the first row, the read and write sequences occur simultaneously. We have added a time arrow to the figure to clarify the sequence.
We are not sure what you mean by the alignment in table 5. We have added a rule between the other methods and our approach.
NL is the number of labels within the architecture. We have reworded the text to make this clearer.
WL is the width (in bits) required to represent the labels.
Reviewer 2 Report
As a paper about connected-component labeling algorithm, the authors should know and introduce the state-of-the-art connected-component labeling algorithms in the field. For example, those algorithms introduced in the following paper. The Connected-Component Labeling Problem: A Review of State-of-the-Art Algorithms. Pattern Recognition, Vol. 70, pp.25-43, 2017.Author Response
We would like to thank the reviewer for bringing this paper to our attention. If focusses primarily on CCL algorithms, with only a couple of paragraphs at the end discussing single-pass CCA. However, we have expanded the introduction with a more complete overview of the range of CCL algorithms, and more discussion of two newer CCA algorithms.
We have also included discussion of the paper by Tang et al. throughout the comparison and discussion section, since we had overlooked that earlier, and it is a relevant alternative approach.
Reviewer 3 Report
In short this is another take on CCA with further improvements. The paper is really of very high quality and rich with analysis and details. I feel that the literature analysis in FPGA implementations has disregarded part of the improvements happened since 2010 when Block Based CCL was introduced (Grana et.al.) . Moreover some of the later improvement by He have further added potential for saving merges in Union Find.
Overall I find the paper really good and ready for publication, but a more in depth analysis of recent CCL literature should be done and possibly included in future works by the authors. Let me suggest a few interesting works:
Grana, C., Borghesani, D., & Cucchiara, R. (2010). Optimized block-based connected components labeling with decision trees. IEEE Transactions on Image Processing, 19(6), 1596-1609.
He, L., Zhao, X., Chao, Y., & Suzuki, K. (2014). Configuration-transition-based connected-component labeling. IEEE Transactions on Image Processing, 23(2), 943-951.
Grana, C., Baraldi, L., & Bolelli, F. (2016, October). Optimized connected components labeling with pixel prediction. In International Conference on Advanced Concepts for Intelligent Vision Systems (pp. 431-440). Springer, Cham.
He, L., Ren, X., Gao, Q., Zhao, X., Yao, B., & Chao, Y. (2017). The connected-component labeling problem: A review of state-of-the-art algorithms. Pattern Recognition, 70, 25-43.
Author Response
We have expanded the introduction with a more complete overview of the range of CCL algorithms (including the suggested papers), and more discussion of two newer CCA algorithms.
We have also included discussion of the paper by Tang et al. throughout the comparison and discussion section, since we had overlooked that earlier, and it is a relevant alternative approach.
Round 2
Reviewer 2 Report
The paper has revised according on my comments.